# Double-Sided Sapphire Optrodes with Conductive Shielding Layers to Reduce Optogenetic Stimulation Artifacts

**DOI:** 10.3390/mi13111836

**Published:** 2022-10-27

**Authors:** Junyu Shen, Yanyan Xu, Zhengwen Xiao, Yuebo Liu, Honghui Liu, Fengge Wang, Chaokun Yan, Liyang Wang, Changhao Chen, Zhisheng Wu, Yang Liu, Peng Un Mak, Mang I. Vai, Sio Hang Pun, Tim C. Lei, Baijun Zhang

**Affiliations:** 1School of Electronics and Information Technology, Sun Yat-sen University, Guangzhou 510006, China; 2State Key Laboratory of Analog and Mixed-Signal VLSI, Institute of Microelectronics, University of Macau, Macau 999078, China; 3State Key Laboratory of Optoelectronic Materials and Technologies, Sun Yat-sen University, Guangzhou 510275, China; 4Department of Electrical and Computer Engineering, Faculty of Science and Technology, University of Macau, Macau 999078, China; 5Department of Electrical Engineering, University of Colorado, Denver, CO 80204, USA

**Keywords:** double-sided optrode, microelectrodes, stimulation artifacts, electromagnetic shielding layers, sapphire substrate

## Abstract

Optrodes, which are single shaft neural probes integrated with microelectrodes and optical light sources, offer a remarkable opportunity to simultaneously record and modulate neural activities using light within an animal’s brain; however, a common problem with optrodes is that stimulation artifacts can be observed in the neural recordings of microelectrodes when the light source on the optrode is activated. These stimulation artifacts are undesirable contaminants, and they cause interpretation complexity when analyzing the recorded neural activities. In this paper, we tried to mitigate the effects of the stimulation artifacts by developing a low-noise, double-sided optrode integrated with multiple Electromagnetic Shielding (EMS) layers. The LED and microelectrodes were constructed separately on the top epitaxial and bottom substrate layers, and EMS layers were used to separate the microelectrodes and LED to reduce signal cross-talks. Compared with conventional single-sided designs, in which the LED and microelectrodes are constructed on the same side, our results indicate that double-sided optrodes can significantly reduce the presence of stimulation artifacts. In addition, the presence of stimulation artifacts can further be reduced by decreasing the voltage difference and increasing the rise/fall time of the driving LED pulsed voltage. With all these strategies, the presence of stimulation artifacts was significantly reduced by ~76%. As well as stimulation suppression, the sapphire substrate also provided strong mechanical stiffness and support to the optrodes, as well as improved electronic stability, thus making the double-sided sapphire optrodes highly suitable for optogenetic neuroscience research on animal models.

## 1. Introduction

Optogenetics is a biochemical technique in which the DNA of light-sensitive ion-channels (opsins) are transfected to neurons, and the action potential firing of the transfected neurons can be excited or inhibited, depending on the type of opsin used, through light stimulation. Optogenetics allows researchers to modulate one or multiple neural targets in the brain with light stimulation without causing damage to the brain tissue; therefore, optogenetics has become an important modality to study neuronal circuits and to seek treatments for neurological disorders, including Parkinson’s disease, depression, and epileptic seizures [1,2,3]. In order to make optogenetics more accessible, the development of “optrodes”—neural probes integrating microelectrodes and light sources—are crucial to simultaneously record neural activity and to give accurate spatial and temporal optical modulations to neurons [1,4,5,6,7]. Several research groups have worked on developing better optrodes, but most of these optrodes are based on silicon substrates, which are very brittle and extremely difficult to handle for actual insertion into animal brains [8,9,10]. Moreover, optrodes developed by the research group at the University of Michigan integrated several planar microelectrodes and LEDs onto the same probe, and these planar optrodes have been used in several optogenetic experiments [1,5]. Kim et. al. also proposed an optrode integrated with LEDs that had a low number of stimulation artifacts [11]. This type of integrated planar optrode has several advantages over traditional fiber-based waveguide optrodes, including higher optical efficiency and minimal probe volume, and it may also lead to lower insertion damage to the brain tissue. In addition, a higher spatiotemporal resolution, both in terms of the optical stimulation and neural recording, can be achieved due to the maturity of the semiconductor technology being used [8,11,12,13,14].

There are some common issues with these types of optrodes. When the LEDs of the optrodes are turned on, stimulation artifacts can be observed from the microelectrodes. These stimulation artifacts can significantly hamper the quality of the neural recordings, especially during the onset of voltage transients. The electrical contaminations can be caused by electromagnetic interferences (EMIs) that are caused by the LED driving currents to the microelectrodes [8,11,15]. Moreover, in such cases, the interfering mechanisms of the optical artifacts fail to correspond with one another, and it has been argued that these optical artifacts are caused by the photovoltaic effect [16,17], photoelectrochemical effect [16,18,19], and photothermal effect [16].

There are some efforts which have been undertaken to reduce the effects of stimulation artifacts on optrodes. Electromagnetic shielding (EMS) has been demonstrated to be an effective way to reduce the presence of electrical artifacts [8,11,15,20,21,22]. In addition, modifying the material surface property or choosing an optimal microelectrode material can also be used to further reduce the presence of stimulation artifacts. Modifying the microelectrode surface by electroplating it with pulsed voltages [23], or coating it with conductive polymers [24], has also been demonstrated as a method with which to reduce the presence of stimulation artifacts. Microelectrodes can also be replaced by some other materials, such as metals with a lower impedance and higher stability [19], transparent materials such as indium tin oxide (ITO), as well as graphene [17,25,26].

In this paper, a double-sided sapphire optrode integrated with a LED for optogenetic stimulation, and an array of sixteen microelectrodes for action potential recording, was developed. The LED and the microelectrode array were constructed separately on the top epitaxial and bottom substrate layers to reduce EMI. Three electromagnetic shielding (EMS) layers were also integrated into the optrode to help shield the microelectrodes from the LED EMIs. Compared with other optrodes, which integrate the LED and microelectrodes on the same side, our double-sided sapphire optrode evidently reduces the presence of electrical artifacts. As well as incorporating EMS multi-layer structures to reduce EMIs, further EMI reduction can be achieved by controlling the onset of the LED driving voltage so that it has a more gradual profile, both in terms of voltage difference and temporal transients. This double-sided optrode design is a new approach for optrode production, and it may bring new benefits to neuroscience studies.

## 2. Experiments and Methods

### 2.1. Device Design

The optrode developed in this work was based on a InGaN/GaN blue-LED structure grown on a sapphire substrate. There are several advantages of using sapphire as the optrode substrate over the more commonly used silicon substrate: (1) sapphire has a significantly greater material stiffness and is much harder to break when implanting the optrode into the animal’s brain; (2) sapphire is optically transparent, thus allowing the light from the LED to illuminate both sides of the optrode; and (3) it is also possible to perform flip-chip bonding (FCB) on a sapphire substrate so that both sides of the substrate are utilized separately for integrating the microelectrodes and LED, on opposite sides, in order to reduce signal interference.

Figure 1a,b illustrates the top epitaxial and bottom substrate surfaces of the double-sided optrode, respectively. The top epitaxial surface was grown with a LED for optogenetic stimulation, with the LED connected to two long interconnection lines with FCB pads for printed circuit board (PCB) bonding. On the bottom substrate of the sapphire optrode, 16 electrophysiology microelectrodes with long interconnection lines and gold (Au) wire-bonding pads were patterned in an interconnecting diamond shape, which was intended for recording action potential (AP) neural activities. A distributed Bragg reflector (DBR) was constructed on the epitaxial side of the optrode to help reflect the light that was generated within the active layer of the LED back to the substrate side which contained the microelectrodes. Since light should be illuminating neurons on the same side that the microelectrodes are recording, this DBR reflection is necessary so that it can reflect light back to stimulate neurons, using the neural recording from the microelectrodes.

The overall dimensions of the double-sided sapphire optrode were measured to be 13 × 0.4 × 0.2 mm^3^, and the width and thickness of the sapphire optrode was minimized to avoid insertion damage to the brain tissue. As shown in Figure 1a, L and W is the length and width of the optrode. Although the entire length of the optrode is L = 13 mm, there is an 8 mm section reserved for tissue insertion in rodent brains. This insertion section can be further extended in the future to accommodate for larger animal brains. To reduce damage to the brain tissue in optogenetic experiments, the flat tip of the optrode in Figure 1 could be processed to form a V-tip shape using picosecond laser and femtosecond laser in the future [27,28].

One innovation of the double-sided sapphire optrode is the integration of three EMS shielding layers to reduce optogenetic artifacts during optical illumination. These electrical artifacts can significantly contaminate the simultaneously recorded neural signals. Figure 1c illustrates the three EMS layers embedded within the optrode in the cross-sectional direction. The first EMS layer was constructed with ITO on top of the *n*-GaN layer, and the LED and the recording microelectrodes were electrically separated. The transparency of ITO allowed the light to illuminate both sides of the probe. The second and third EMS layers were constructed with Cr/Au on the outermost surfaces of both the epitaxial and substrate layers. EMS layer Ⅱ reflected the light from the LED’s side to the microelectrodes’ side, and a window was reserved on EMS layer Ⅲ, without depositing metal, to make sure that light was only emitted around the microelectrodes and no other optical leakage occurred. These three EMS layers surrounded the optrode entirely to shield environmental interferences and separate the electrical input from active devices during neural recordings.

Figure 1d is a photograph showing the optrode mounted on a custom PCB to allow interfacing to neural amplifiers and LED-driven electronic devices. The photograph was also taken with a measurement ruler to show that the entire optrode, with the PCB, has an overall length of ~2 cm. Figure 1e shows the epitaxial side of the optrode and the outline of the LED is clearly visible. Figure 1f,g shows the substrate side of the optrode with the neural recording microelectrodes when the LED is turned off (f) and turned on (g). Figure 1h is a micrograph showing the back side of the optrode bonded to the PCB conducting pads.

### 2.2. Fabrication and Encapsulation

GaN-based LED technology, on a sapphire substrate, was used to fabricate our double-sided sapphire optrode embedded with EMS layers. LED epitaxial layers were grown on the sapphire substrate using metal organic chemical vapor deposition (MOCVD). The LED epitaxial layers included a GaN buffer layer, an *n*-GaN layer, GaN/InGaN MQW for the LED active region, an AlGaN electron blocking layer, and a *p*-GaN layer, as shown in Figure 2a.

The detailed procedure for manufacturing the optrode can be divided into three steps—processing the epitaxial layers, processing the sapphire substrate, and encapsulating the optrode.

To process the LED on the top epitaxial layer, a 400 nm deep mesa was etched onto the *n*-GaN layer using inductive coupled plasma (ICP), followed by the activation of the *p*-GaN layer at 830 °C for 30 min in an alloy furnace, as illustrated in Figure 2a–g. An ITO layer was then grown on the *p*-GaN layer through sputtering, and it was rapidly thermally annealed for 5 min at 570 °C to form an ohmic contact with the *p*-GaN layer. The ITO layer was then the first EMS layer to be electrically separated from the LED and the microelectrodes. A 1-μm-thick SiO_2_ layer was then deposited onto the ITO to form an isolation layer for the long interconnection wires using plasma-enhanced chemical vapor deposition (PECVD). A metal layer of chromium/titanium (Cr/Ti) was deposited to create ohmic contacts with the *n*-GaN layer, and this metal layer was also used as the anode, the cathode, and the long interconnection wires for the LED. A SiO_2_/TiO_2_ DBR was also deposited on top of the metal layer to reflect light back onto the sapphire surface. Window pads that were used to connect the Cr/Ti metal layer were opened with ICP dry etching on the DBR layer, and a layer of Cr/Au was deposited onto the DBR layer to form FCB pads so that the LED could connect to the Cr/Ti layer for electrical conduction. This Cr/Au layer was connected to the cathode and n-electrode pads, and it also served as the second EMS layer to cover the LED structure. A SiO_2_ passivation layer was finally deposited with PECVD to cover the entire LED structure, but connection windows for the LED pads were left open to allow FCB to occur via wet etching.

The process of constructing the microelectrodes on the sapphire substrate is illustrated in Figure 2h–l. Back grinding and chemical mechanical polishing (CMP) were performed to prepare the back surface of the sapphire substrate for the construction of the microelectrodes. A gold (Au) layer was deposited in order to construct the microelectrodes and their interconnection wires. An isolation layer of SiO_2_ was deposited by PECVD, and it was wet etched to create openings for the microelectrodes and wire-bonding pads. A third EMS layer, and pads for the microelectrodes, were manufactured by depositing another Cr/Au metal layer on top of the SiO_2_ insulation layer. Care was taken not to deposit the Cr/Au metal onto the LED, thus ensuring that light would not be blocked and could reach the top from the substrate underneath. Finally, a passivation layer was deposited by PECVD, excluding the microelectrode pads, to isolate the microelectrode.

After both sides of the sapphire wafer had been processed, the sapphire wafer was sawed to obtain individual optrodes. The LED pads on the epitaxial layer of the probe were flip-chip bonded onto an interfacing PCB with solder paste, as illustrated in Figure 2m. Microelectrode pads on the epitaxial layer were also wire-bonded to the PCB with aluminum wires, as shown in Figure 2n. Epoxy glue was applied onto the bonding area to provide mechanical protection and electrical isolation, in order to protect the aluminum microwires. The PCB bonding to the sapphire probe was necessary to provide an easy electrical connection to the neural amplifiers and LED drivers, and it also provided a mounting platform to hold the sapphire optrode for brain insertion to the animal. A connection block was also soldered onto the PCB to provide it with standard connectivity and interfacing. The final double-sided sapphire optrode, integrated with 3 EMS layers, is illustrated in Figure 2o.

## 3. Results and Discussions

### 3.1. Electrical and Optical Performance

After fabricating the double-sided sapphire optrodes, the electrical and optical properties of the optrode were measured and characterized. The current-voltage (I_dc_-V_dc_) characteristics of the LED were measured using a semiconductor device parameter analyzer (Agilent B1500A, Keysight, Penang, Malaysia). The I_dc_-V_dc_ curves, measured with and without the PCB, were compared in Figure 3a. For both cases, the turn-on voltage of LED was measured to be ~2.54 V with I_dc_ = 1 mA. The series resistance of the optrode was measured to be 30.75 Ω with the FCB pads, and it increased to 32.09 Ω without the FCB pads. This increase in resistance was likely due to the increase in contact resistance between the parameter analyzer and the optrodes without the FCB pads; therefore, it was not related to FCB pads. Since no additional resistance was found, an increase in temperature should not be a concern for the FCB.

The stability of the optrodes was evaluated by submerging the optrode in a 0.9% phosphate buffer solution (PBS) for an extended period. PBS has a similar osmolality to body fluid, with a PH value of ~7.1, and it is commonly used to simulate biological environments [11,17]. The sapphire optrode was immersed in PBS for 0, 6, 12, and 24 h, respectively. The measured I_dc_-V_dc_ curves of the optrode, after different immersion times, are shown in Figure 3b. The I_dc_-V_dc_ curves remained constant as the immersion time increased, which indicates that there was no degradation to the optrode after an extended period in the solution. Our optrode showed excellent reliability in the biological environment, and it satisfied the experimental time of the optogenetic experiment.

The electroluminescence (EL) spectrum and the optical power density of the blue LED were measured and plotted in Figure 3c,d, respectively. The EL spectrum was measured with an integrating sphere equipped with a highly accurate array spectroradiometer (HAAS-2000, Everfine Corporate, Hangzhou, China), it has a peak wavelength of 444.5 nm, and a FWHM bandwidth of 20 nm at 10 mA driving current. The optical power density of the blue LED was measured with an optical power meter (843-R, Newport Corporate, Irvine, CA, USA). The optical power density increased linearly for I_dc_ < ~15 mA, and it became saturated as the I_dc_ continued to increase; again, this was likely due to local heating and carrier overflow. Generally speaking, the activation threshold of channelrhodopsin-2 (ChR2) in animal brains is estimated to be ~1 mW/mm^2^ [12], and it can be comfortably reached by the LED with a moderate driving current of ~6 mA. Typical optogenetic neural stimulation will not allow the LED to emit light for a long period of time; instead, pulsing the LED to initiate stimulations or to inhibit action potentials is a common experimental strategy to avoid other detrimental effects to the neurons.

The impedance of the microelectrode/electrolyte interface was also measured, as shown in Figure 3e. Electrochemical impedance spectroscopy (EIS) of a microelectrode was measured in PBS with an LCR impedance meter (E4980A, Keysight, Santa Rosa, CA, USA) with a 50 mVpp AC driving voltage and no reference bias. The measured impedance of the microelectrode decreased exponentially as the driving frequency continued to increase, and as the microelectrode dimension became larger. More specifically, the impedance values measured at the 1 kHz driving frequency were 3.75, 2.92, 2.21, and 1.52 MΩ for microelectrodes with a diameter of 10, 15, 20, and 25 μm, respectively. Measuring impedance at 1 kHz is considered to be the standard for neural recording electrodes since the pulse width of an action potential is 1 ms [29].

### 3.2. Reduction of Optogenetic Stimulation Artifacts

Since the introduction of the first optrode, an undesirable issue which significantly hampers the recorded neural signal quality, is the presence of stimulation artifacts [8,11,15,16,17,18,19]. A stimulation artifact may be defined as either an electrical or optical artifact. An electrical artifact may emerge due to the proximity of a LED to the microelectrodes. The various, rapid driving voltages of the LEDs can result in significant EMIs, and these EMIs can couple with the microelectrodes, thus resulting in noise contamination; however, the origin and physical mechanisms causing these optical artifacts are unclear [16,17,18,19].

In this paper, the stimulation artifacts of the double-sided sapphire optrodes were investigated in detail, and a testing system was developed to investigate these noise artifacts, as shown in Figure 4a. In this testing setup, the active area of the optrode was submerged in PBS, and a copper surrounding was used to shield EMIs generated by other instruments from the optrode. One of the microelectrodes of the optrode was connected to an amplifier powered by a DC voltage source (LPS-305, Motech Electronic, Ningbo, China). The amplifier output was then sent to a data acquisition card (USB-6215, National Instruments, Debrecen, Hungary), or a digital storage oscilloscope (DSOX4024A, Keysight, Penang, Malaysia), and then to a computer for data acquisition [30]. Moreover, the anode and cathode of the LED, on the same optrode, were connected to a pulsed signal generator (DG4062, Rigol, Suzhou, China) in order to supply pulsing voltages to drive the LED. In addition, the instrument grounds, the three EMS layers, the cathode of the LED, the copper EMI shield, and the submerged area in the PBS were all connected together on the ground to avoid ground loops.

Figure 4b plots the pulsed LED voltages, with a baseline voltage maintained at zero, and different driving voltages ranging from 2.5 V to 2.7 V. The repetition rate and duty cycle of the pulsed voltages were set at 10 Hz and 50%, respectively. For comparison purposes, we also constructed both single-sided and double-sided optrodes without EMS layers to compare with the double-sided optrodes that were embedded with the EMS layers.

Stimulation artifacts were more pronounced for the single-sided microelectrodes, as shown in Figure 4c. With the same driving voltages, the double-sided optrode had a significantly fewer stimulation artifacts than the single-sided electrode. The amplitudes of stimulation artifacts declined to 52.46%, 60.81%, and 56.62% under the driving voltages of 2.5 V, 2.6 V, and 2.7 V, respectively. Roughly speaking, compared with single-sided optrodes, the number of artifacts was reduced by half when using the double sided optrode. For single-sided optrodes, the interconnection lines carry both the LED driving voltages and the recorded neural signals that were constructed on the same metal layer and packed very densely; these closely spaced interconnection lines can couple together, thus leading to significant signal contamination. In contrast, double-sided optrodes separate the interconnection lines that drive the LED from those lines that are connected to the microelectrodes, which, in turn, reduces EMIs effectively. The use of the *n*-GaN layer as an EMS separation layer between the microelectrodes and the LED could further protect the microelectrodes from signal contamination due to optical stimulations.

The further reduction of artifacts can be achieved by shielding the optrodes from the environment with EMS layers grown on the top and bottom superficial layers. As depicted in Figure 1c, three conductive films were deposited on the *n*-GaN layer, the LED structure, and the microelectrodes. These conductive films were deposited with the ohmic contacts of the *p*-GaN layer, the LED pads, and the microelectrode pads, such that the construction steps can be simplified. The first EMS layer was sandwiched between the LED and microelectrodes, which served similar roles to the *n*-GaN layer in terms of shielding. The second EMS layer covered the LED structure from the top superficial layer and was used to shield EMIs generated by the LED driving circuits from leaks. The third EMS layer was deposited onto the microelectrode structures at the bottom to protect the microelectrodes, and the recording lines, from picking up interferences from the LED and other environmental noises.

The number of stimulation artifacts that were measured as a result of the use of double-sided optrodes without (solid lines) and with (dash lines) the three EMS layers are shown in Figure 4d. The stimulation artifacts were markedly reduced when the three EMS layers were present. With three EMS layers, the maximum amplitudes of the stimulation artifacts were reduced to 61.30%, 63.99%, and 88.31% at driving voltages of 2.5 V, 2.6 V, and 2.7 V, respectively. These results strongly support the hypothesis that EMS layers play a critical role in reducing the presence of stimulation artifacts. There were still some remaining artifacts, even with the three EMS layer design, which may be ascribed to other optical and residual electrical artifacts. Due to technical limits in the deposition process, the optrode sidewalls were not covered by conductive films, and the small window on the mesa was opened to allow light emissions to pass through. These areas were not shielded, and may cause EMI leaks, thus contributing to other residual electrical artifacts.

These measured results may also indicate voltage transients at the onset of optical stimulation, which may result in the presence of a significant number of artifacts. It is apparent from Figure 4c,d that at the onset, the higher the voltage difference between the baseline voltage and the driving voltage, the greater the number of stimulation artifacts. This hypothesis can be tested by raising the baseline voltage from zero to slightly below the LED emission threshold in order to reduce the number of voltage transients at the onset. Figure 5a shows a series of LED driving voltages; the driving voltage was kept at 2.7 V, but the baseline voltages increased from 0 to 2.2 V, which were still below the LED light emission threshold. This approach guaranteed that the optical power density emitted from the LED remained unchanged since light was only being emitted while the higher voltage was applied; however, it did not emit light while the baseline voltage was below the LED emission turn-on voltage. Figure 5c shows that the number of stimulation artifacts were significantly reduced as a result of the double-sided optrode, embedded with the EMS multi-layer structure, being used. The peak-to-peak voltage (Vp-p) of the artifacts reduced to 43.44% when the driving voltage was raised to 2.2 V. This illustrates that a smaller transient voltage can further mitigate the effects of stimulation artifacts.

The analysis above also hinted that modifying the slew rate of the voltage transients, during the rise and fall times of the LED stimulation, may also alleviate the EMI effect; therefore, we also tested this hypothesis by increasing the rise and fall times of the transients of the LED voltage. In Figure 5b, black, red, and blue lines were used to plot steep LED driving voltages, from 0 to 2.7 V, and the other three lines were used to plot more gradual LED driving voltages, from 2.2 to 2.7 V. For both transients, rise/fall times of 0.2, 0.6, and 1 ms were used to investigate the stimulation artifacts. Figure 5d clearly demonstrates that the amplitudes of the stimulation artifacts can be further reduced with a longer rise time for both transients. Due to the rise time increasing, the peaks of the stimulation artifacts also lagged at the onset, but the recovery time of the stimulation artifacts still remained unchanged for ~6 s; therefore, extending the delay time further mitigated the electrical artifacts without increasing the recovery time by a significant amount.

## 4. Conclusions

In this work, a low-noise double-sided sapphire LED optrode, integrated with multiple EMS layers, was developed. GaN on a sapphire substrate provides much-improved mechanical stiffness, as compared with other substrate materials such as silicon, for the optrode, thus making it less susceptible to breaking during its insertion into the brain tissue. The sapphire optrodes are also electrically and optically stable in biological solutions. In addition, the transparent nature of sapphire allows light to be emitted in the substrate layer, where the microelectrodes are, thus allowing the LED and the microelectrodes to be constructed on different sides of the optrode in order to avoid signal cross-talks. In this work, a double-sided sapphire optrode was designed and constructed. The LED and the microelectrodes were made separately on the top epitaxial and bottom substrate layers of the optrode, and it was electrically isolated by inserting an ITO EMS layer. In addition, the middle shielding layer also avoided the long signal interconnection lines, which were contaminated with EMIs, and which were also induced during optical stimulation. Our results indicated that the stimulation artifacts produced by the double-sided sapphire optrodes can be reduced by half with this novel strategy. Further EMI reduction can be achieved by growing two EMS layers on the top and bottom surfaces to prevent noise from leaking; this noise is generated by the LED driving circuits, and it can contaminate microelectrode recordings. Another strategy to further reduce the presence of stimulation artifacts is to reduce the number of LED driving voltage transients by increasing the baseline voltage, ensuring that it is below the LED threshold voltage, and by using a more gradual rise and fall time profile to avoid large voltage transients. Based on these strategies, the number of stimulation artifacts can be reduced by as much as 76%.

## Figures and Tables

**Figure 1 micromachines-13-01836-f001:**
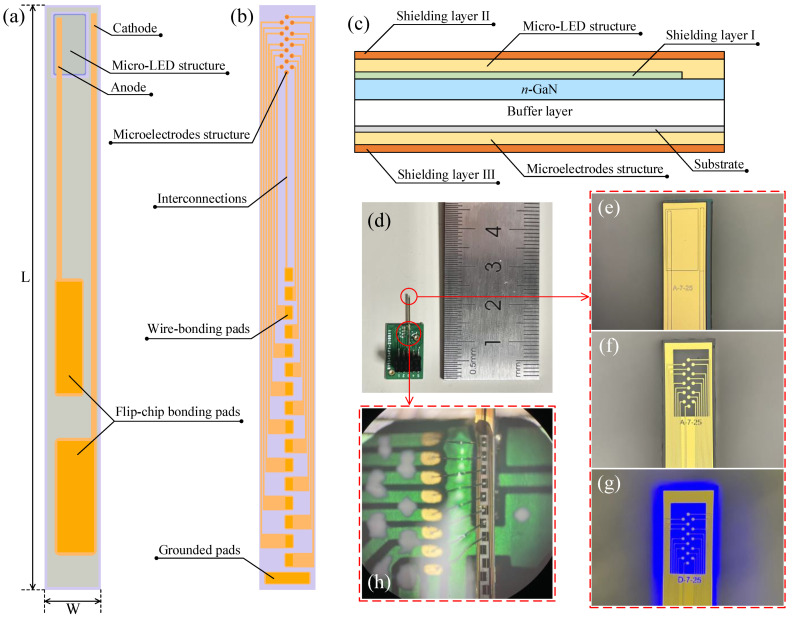
(**a**) The top epitaxial layer constructed with a LED and (**b**) the bottom substrate layer with the multiple microelectrodes of the double-sided optrode; (**c**) the cross-sectional view of the optrode showing the EMS multi-layer structure with shielding layers that can separate the microelectrodes and the LED to reduce the presence of optogenetic stimulation artifacts; (**d**) a photograph of the optrode mounted onto a PCB (green); (**e**–**g**) micrographs of (**e**) the epitaxial LED (top side), (**f**) the microelectrodes on the substrate (back side), and (**g**) the optrode when the blue LED is emitting light; and (**h**) a micrograph shows the back side of the optrode (grey, center) bonded to the PCB conducting pads.

**Figure 2 micromachines-13-01836-f002:**
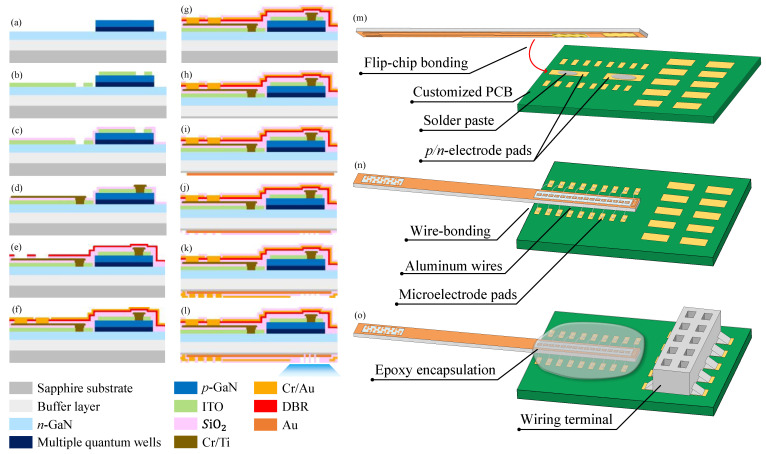
The microelectronic processing steps for constructing and encapsulating a double-sided sapphire optrode with three EMS layers to minimize the presence of optogenetic electrical artifacts during recording. (**a**) Mesa etching, (**b**) ohmic contact with the *p*-GaN and the first EMS layers, (**c**) isolation layer deposition, (**d**) ohmic contact with the *n*-GaN, (**e**) DBR deposition for light reflection, (**f**) Cr/Au deposition for *p*/*n*-electrode pads and the second EMS layer on top, (**g**) passivation layer deposition for epitaxial layer insulation, (**h**) back surface grinding and CMP for surface preparation, (**i**) Cr/Au deposition to create the microelectrodes and interconnection lines, (**j**) deposition of the SiO_2_ isolation layer, (**k**) Cr/Au deposition to fill the microelectrode pads and form the third EMS layer, (**l**) SiO_2_ passivation layer on the substrate, (**m**) flip-chip bonding for connecting the LED pads to the PCB, (**n**) wire-bonding to connect the microelectrode pads to the PCB, and (**o**) epoxy encapsulation to protect the bonded area and add a wiring terminal for easy interfacing.

**Figure 3 micromachines-13-01836-f003:**
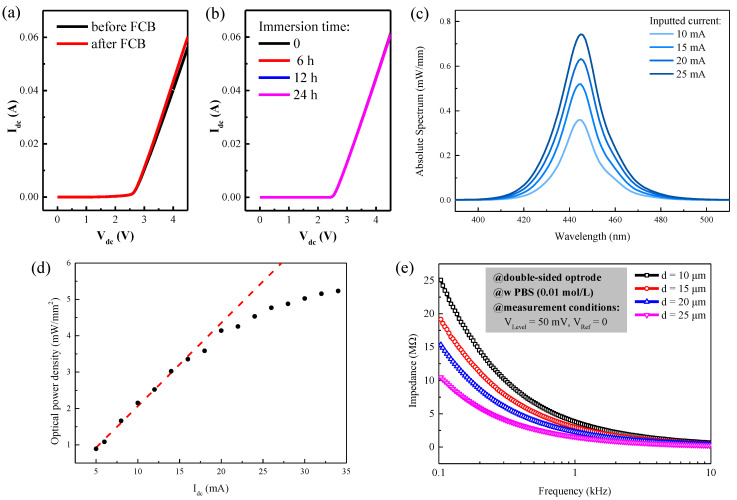
(**a**) The I_dc_-V_dc_ curves with and without flip-chip bonding. (**b**) The I_dc_-V_dc_ curves of an encapsulated optrode measured before and after immersion in PBS. (**c**) Electroluminescence (EL) spectrum of the LED on the optrode, measured with increasing LED driving currents of 10, 15, 20, and 25 mA. (**d**) The measured curve of optical power density versus input current I_dc_. (**e**) Electrochemical impedance spectroscopy (EIS) of microelectrodes with different radii of 10, 15, 20, and 25 μm.

**Figure 4 micromachines-13-01836-f004:**
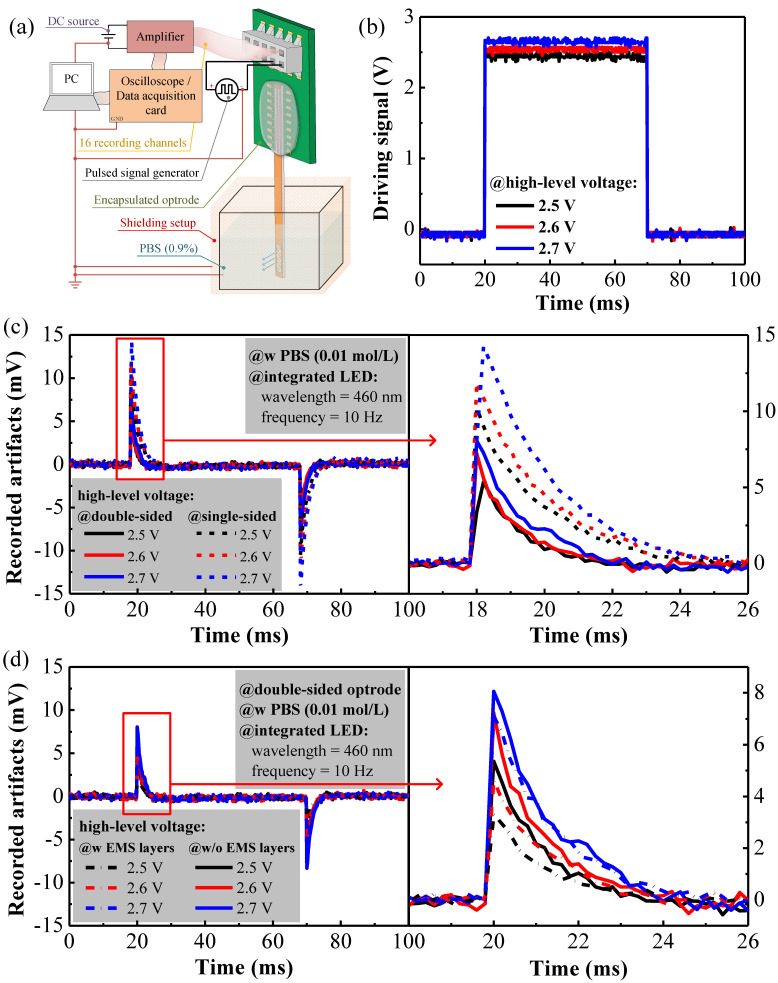
(**a**) The testing system for the stimulation artifacts. (**b**) The pulsed LED driving voltage of 2.5, 2.6, and 2.7 V; the baseline voltage maintained at zero. (**c**) Stimulation artifacts under different driving voltages measured on a single-sided optrode and on a double-sided optrode. One of the artifact profiles is enlarged for better readability. (**d**) Stimulation artifacts under different driving voltages measured on a double-sided optrode with and without the three EMS layers.

**Figure 5 micromachines-13-01836-f005:**
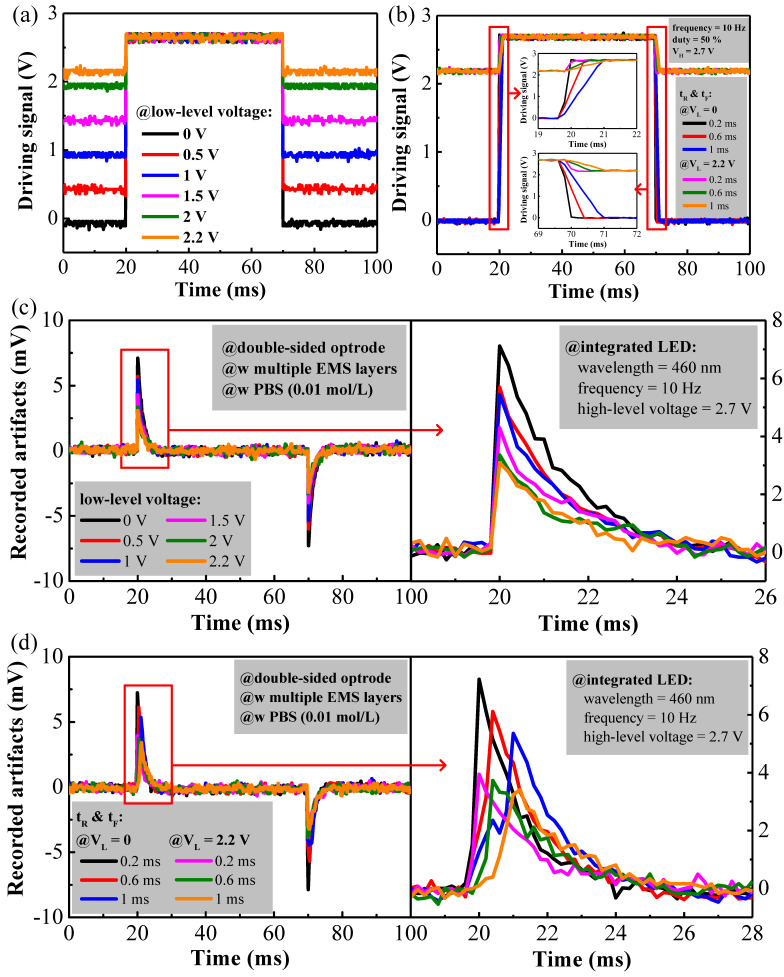
(**a**) The pulsed LED driving voltage was maintained at 2.7 V while the baseline voltage was 0, 1, 1.5, 2, and 2.2 V below the LED emission threshold. (**b**) The pulsed LED driving voltage with different rise/fall times of 0.2, 0.6, and 1 ms. (**c**) Stimulation artifacts under different driving voltages, as shown in (**a**). One of the two artifacts were enlarged on the right for better readability. (**d**) Stimulation artifacts under the different driving voltages shown in (**a**), and with the different rise/fall times shown in (**b**).

## Data Availability

The data presented in this study are available on request from the corresponding author. The data are not publicly available due to relative proceeding experiments.

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
