# Peer review of "Double-Sided Sapphire Optrodes with Conductive Shielding Layers to Reduce Optogenetic Stimulation Artifacts"

_micromachines, 2022, doi:10.3390/mi13111836_

Round 1

Reviewer 1 Report

Prof. Zhang and colleagues present a creative idea: using EMS isolation layer in conjunction with Double sided geometry of LED and electrodes to reduce interference from LEDs for optogenetic to electrophysiology recording electrodes. The quality of the device and details of process generating such device is well suitable for Micromachines. However, for several reasons, the current manuscript has failed to demonstrate its benefit. Therefore I strongly suggest authors to revise the manuscript to answer below concerns,

1)    Electromagnetic shielding (EMS) seems little miss leading. How does the shielding layer (1~3) affect optical reflection?

2)    Electromagnetic shielding wise, how much does the shield? What impedance do we measure on LED drive – to – electrode? I suspect, little bit of capacitive coupling and little bit of mutual inductance…

3)    From the statement “With all these strategies, the 33 stimulation artifacts were significantly reduced by ~76%” please separate how much each approach - EMS and amplitude (slope) reduces the “crosstalk”

4)    Optogenetics technically does not turn on or off rather modulates the behavior of a targeting neuron please correct.

5)    It is unclear what three shielding layers of the proposed work intends to remove from the broad list of suggested sources of artifact including EMI, photovoltaic, photoelectrochemical, photothermal. It reads like its’ only concerning EMI.

6)    If EMI is the major concern, why does it matter to measure artifact in buffer (water) vs. air? If just a different dielectric constants does it match the bulk body calculation?

7)    One of the key idea of this work is to use different side of the shank for each purpose.

A.    What example of circuit allows study of circuit that is separated by the shank?

B.     How much light do you expect to be scattered in the brain tissue and how much do you expect it to arrive at the electrodes?

8)    Amplitude of LED driving voltage (which decides current and amount of light) should be set enough to modulate cells instead of lowering interference?

9)    There seems to be little more concern of coupling at the PCB, connector, wires.

10) Please comment on how does this compare to other work that uses signal processing (circuits go generate less high frequency component on LED driving pulse) approaches in the literature.

Most importantly, please support the mechanism of how each suggested technique can contribute (quantitatively) to reduce the artifact, instead of just showing numbers. Also the measurement setup seems quite critical to highlight the result. Please add details.

Author Response

Thank you for your review. We were pleased to receive such valuable and constructive comments about our work. We thank the reviewer for the time and effort that they have put into reviewing the previous version of the manuscript. The suggestions have enabled us to improve our work.

Based on the instructions provided in your letter, we uploaded the letter including our point-by-point response to the comments raised by the reviewers. The comments were reproduced and our responses were given directly afterward. We also uploaded the file of revised manuscript with all the changes marked up using “Change Track” function.

We would like also to thank you for allowing us to resubmit a revised copy of the manuscript.

We hope that the revised manuscript is accepted for publication in the Micromachines.

Sincerely,

Junyu Shen

Reviewer 2 Report

This is an interesting paper about the fabrication of a double-sided optrode in a sapphire substrate, integrating a LED, gold microelectrodes, and electromagnetic shielding (EMS) layers to reduce stimulation artifacts. Overall, the paper is well-structured. However, I have some important revisions and comments:

1. An English revision may be done to some sentences along the paper, e. g. in line 26: “In this paper, we tried to mitigate the by developing a low-noise double-sided optrode integrated with multiple Electromagnetic Shielding (EMS) layers.”

2. Concerning the methods section, the major concern is the flat tip of the probe, instead of a vtip shape, which probably will hamper its future insertion in the animal brain and may increase brain tissue damage. Additionally, optrode width (400um) and thickness (200um) are a bit high.

3. Also in the methods section, why use a different material for EMS layer 1 (ITO) and EMS layers 2 and 3 (Cr/Au)?

4. In results section 3.1, Figure 3 (d) should be labeled with the meaning of each curve, as made in (a), (b), (c), and (e).

5. Concerning the impedance values obtained at 1kHz, they are higher (> 1Mohm) than the expected impedance values for gold microelectrodes with a radius higher than 10um. These high impedance values will probably hamper the neural recordings.

6. Were the stimulation artifacts tests (results section 3.2) performed only for one microelectrode of the optrode? Will the choice of this microelectrode (in terms of its localization at the optrode) influence the obtained results?

In summary, this paper presents an interesting approach to reducing stimulation artifacts in optrodes, but some major issues/choices should be explained along the paper. Moreover, the future test of this optrode in rodent brains will be hampered by some crucial issues, especially probe geometry (flat tip) and microelectrodes impedance.

Author Response

(The authors gave the same response as above.)

Reviewer 3 Report

The authors present an interesting work for the special topic in Double-sided sapphire optrodes with conductive shielding layers to reduce optogenetic stimulation artifacts. The paper is presents an interesting investigation and written reasonably well. There are still some deficiencies and doubts need authors to dispel. A minor revision is required as indicated in comments to authors.

Specific comments:

1.     Line 223. “The sapphire optrode was immersed in PBS for 30 minutes …”. In the author's experiment to detect the stability of the optrode, the test time is too short. Usually the optrode detection time will not only take 30 minutes to detect. Because the operation process and experimental time of the entire biological experiment are often more than 30 minutes. And the optrode is likely to be reused. Therefore, it is recommended to lengthen the detection time.

2.     In Figure 4(a), the author describes the structure of the whole detection experiment. In the outer periphery of the detection sample (PBS), an extra copper EMI shielding is added to prevent the noise interference generated by the instrument. However, in actual biological experiments, it is difficult to set up copper EMI shielding. Therefore, please explain the effect on reduction of optogenetic stimulation artifacts without this copper EMI shielding.

Author Response

Response to reviewer 3

Thank you for your review. We were pleased to receive such valuable and constructive comments about our work. We thank the reviewer for the time and effort that you have put into reviewing the previous version of the manuscript. The suggestions have enabled us to improve our work.

Based on the instructions provided in your letter, we uploaded the letter including our point-by-point response to the comments raised by the reviewers. The comments are reproduced and our responses are given directly afterward. We also uploaded the file of revised manuscript with all the changes marked up using “Change Track” function.

We would like also to thank you for allowing us to resubmit a revised copy of the manuscript.

We hope that the revised manuscript is accepted for publication in the Micromachines.

Sincerely,

Junyu Shen

Round 2

Reviewer 2 Report

As I previously mentioned, this is an interesting paper showing an approach to reducing stimulation artifacts in optrodes, which is an important topic in optogenetics. However, I cannot accept this paper in its current form due to two main issues:

1.         The authors did not present a justification for the obtained high impedance values (e. g. fabrication process). Since they have knowledge about surface optimization (as they referred to in the cover letter), they should have performed a surface modification to the gold microelectrodes to reduce the impedance.

2.         I believe that it is incorrect to affirm that: “Yes, this is a drawback of sapphire-based microprobes, which cannot be processed into a type V-tip like silicon-based probes...” There are sapphire optrodes reported in the literature with a V-tip shape (e.g. DOI: 10.1109/EMBC.2016.7592141).

Author Response

Thank you for your review. We were pleased to receive such valuable and constructive comments about our work. We thank the reviewer for the time and effort that you have put into reviewing the previous version of the manuscript. The suggestions have enabled us to improve our work.

Based on the instructions provided in your letter, we uploaded the letter including our point-by-point responses to the comments raised by the reviewers. The comments are reproduced and our responses are given directly afterward. We also uploaded the file of revised manuscript with all the changes marked up using “Change Track” function.

We would like also to thank you for allowing us to resubmit a revised copy of the manuscript.

We hope that the revised manuscript is accepted for publication in the Micromachines.

Sincerely,

Junyu Shen

Round 3

Reviewer 2 Report

The authors have presented the justifications and possible solutions for the two main issues that I have presented in the last revision. I accept their answers. Minor english revisions can be perform before publication.